# Effective and Easy Techniques of Collagen Deposition onto Polylactide Films: DC-Discharge Plasma Treatment vs. Chemical Entrapment

**DOI:** 10.3390/polym14224886

**Published:** 2022-11-12

**Authors:** Tatiana S. Demina, Mikhail S. Piskarev, Aisylu V. Birdibekova, Nadezhda N. Veryasova, Anastasia I. Shpichka, Nastasia V. Kosheleva, Andrey K. Gatin, Elena A. Skryleva, Elena V. Istranova, Alla B. Gilman, Tatiana A. Akopova, Peter S. Timashev

**Affiliations:** 1World-Class Research Center “Digital Biodesign and Personalized Healthcare”, Sechenov University, 8-2 Trubetskaya Str., 119991 Moscow, Russia; 2Enikolopov Institute of Synthetic Polymer Materials, Russian Academy of Sciences, 70 Profsoyuznaya Str., 117393 Moscow, Russia; 3Institute for Regenerative Medicine, Sechenov University, 8-2 Trubetskaya Str., 119991 Moscow, Russia; 4Chemistry Department, Lomonosov Moscow State University, 1-3 Leninskie Gory Str., 119991 Moscow, Russia; 5FSBSI Institute of General Pathology and Pathophysiology, 8 Baltiyskaya, 125315 Moscow, Russia; 6N.N. Semenov Federal Research Center for Chemical Physics, Russian Academy of Sciences, 4/1 Kosygina Str., 119991 Moscow, Russia; 7Department of Materials Science of Semiconductors and Dielectrics, National University of Science and Technology “MISiS”, 4 Leninskiy pr., 119049 Moscow, Russia

**Keywords:** polylactide, collagen, surface modification, plasma treatment, biocompatibility

## Abstract

Enhancement of cell adhesion and growth on surface of the biodegradable materials is one of the important tasks in development of materials for regenerative medicine. This work focuses on comparison of various methods of collagen coating deposition onto polylactide films, aiming to increase their biocompatibility with human mesenchymal stromal cells. The collagen deposition was realized using either preliminary plasma treatment of the polylactide films or pre-swelling in solvent mixture. These techniques were compared in terms of the effect on the surface’s chemical structure, morphology, hydrophilicity and ability to support adhesion and growth of human mesenchymal stromal cells.

## 1. Introduction

Immobilization of biomacromolecules onto a synthetic polymer material’s surface is one of the important aspects of development of scaffolds for tissue engineering [1]. Biodegradable polyesters, such as polylactide, are approved by the Food and Drug Administration and have a long-standing history of application in medicine; however, due to their hydrophobicity and a lack of specific sites for cell adhesion, the application of polyester-based scaffolds for tissue engineering is limited. Modification of the polylactide-based material’s surface is widely used to enhance animal cell adhesion and to successfully integrate the scaffold into surrounding living tissues [2,3,4].

A considerable number of works deal with immobilization of biopolymers, i.e., polysaccharides (hyaluronic acid, chitosan, etc.) and proteins (collagen, gelatin, etc.) onto polylactide material surfaces. Due to the hydrophobicity and inert nature of the polylactide substrate, the process of immobilization or coating deposition could be a rather tricky one and require the application of complicated techniques or harsh reagents [5,6]. Immobilization or grafting of bioactive components normally requires preliminary surface activation/functionalization using a hydrogen peroxide solution under UV irradiation or 1,6-hexanediamine–propanol solution followed by application of cross-linking agents, such as glutaraldehyde or methacrylic acid, to graft a range of biopolymers [7,8]. Solvent/non-solvent treatment (also the so-called chemical entrapment method) based on pre-swelling of the polymeric substrate surface layer was shown to be a simple and effective approach for immobilization of bioactive components [5,9,10]. Another popular eco-friendly technique of surface modification is plasma treatment, which allows generation of function groups for future biomacromolecule adsorption/grafting [11]. Plasma-treated substrate could be immediately immersed into a solution of targeted biomacromolecules to adsorb them onto its surface or pretreated with additional chemical agents to enhance a covalent grafting of biocomponents [7,12,13,14]. Plasma treatment is a powerful and flexible tool providing a possibility to regulate morphology and chemistry of the surface layer via variation a number of parameters, such as nature of the working gas, pressure, discharge type, duration of treatment, etc. [12,15]. The plasma polymerization technique could be successfully used to generate amine-containing plasma polymer coatings, which provides better mammalian cell growth on the surface [16]. However, plasma treatment requires rather high-cost equipment and no comparative study of its effectiveness in terms of biomacromolecule immobilization was found to confirm its superiority over more simple chemical entrapment methods.

Thus, in this research we compared the effectiveness of chemical entrapment and plasma treatment approaches to collagen immobilization onto polylactide substrate in terms of generated surface chemical structure, morphology, and the ability to support adhesion and growth of human mesenchymal stromal cells.

## 2. Experimental

### 2.1. Material

Poly(L,L-lactide) (Sigma-Aldrich, Darmstadt, Germany) with an average Mw of 160 kDa was used in this study as received. Collagen (type I) was derived from cattle dermis as reported previously [17]. All solvents were purchased from Chimmed (Moscow, Russia) and were used without further purification.

### 2.2. Preparation of Native and Collagen-Coated PLLA Films

Polylactide films were prepared by casting a 4 wt% solution of poly(L,L-lactide) in CH_2_Cl_2_ onto glass Petri dishes. The films were dried in a dust-free chamber at room temperature. The films formed at the solution/glass interface were used for further study. Coating of PLA films with collagen was carried out using either (1) preliminary plasma treatment or (2) via the chemical entrapment method.

In a frame of the first option, the PLA films were treated by DC-discharge (I = 50 mA) at the anode or at the cathode using filtered air as a working gas (*p* ~ 10 Pa) for 60 s [18]. The films activated by DC-discharge treatments at the anode or the cathode were marked as PLA-A or PLA-C, respectively. Then, the plasma-treated films were incubated using 0.25 wt% collagen solutions in 1% acetic acid for 2 h at 37 °C. The films were carefully washed in bidistillate water and dried in a dust-free chamber. The films treated by plasma at the anode or at the cathode and coated by collagen were assigned as PLA-A-col and PLA-C-col, respectively.

As a second option, the surface of polylactide films was preliminary semi-dissolved in a mixture of acetone/water (7/3 *v/v*) for 1 h, then the film was rapidly transferred into a 0.25% collagen solution and incubated for 2 h. Then, the film was rinsed with distilled water, dried, and marked as PLA-S-col.

### 2.3. Characterization of the Film Samples

Contact angle of wettability (θ) measurements of initial, plasma-treated, and collagen-coated polylactide films were carried out with an Easy Drop DSA100 (KRUSS, Germany) using two test liquids (bidistilled water and glycerol). The values of work of adhesion (W_a_), the total surface energy (γ), and its polar (γ^p^) and dispersion (γ^d^) components were calculated according to [19].

Morphology of the film samples was studied using scanning electron (SEM) and atomic-force (AFM) microscopy with the PhenomProX (Thermo Fisher Scientific, Waltham, MA, USA) and Solver HV (NT-MDT, Moscow, Russia), respectively. The chemical structure of the initial and plasma-treated PLA film surfaces was analyzed by X-ray photoelectron spectroscopy (XPS) using the PHI 5500 VersaProbeII spectrometer, as described previously [12].

A qualitative analysis of collagen immobilization onto polylactide substrates was carried out using fluorescent microscopy (Leica Microsystems, Wetzlar, Germany) of collagen-coated films stained with fluorescein isothiocyanate (FITC), a selective-to-protein label. The native PLA and collagen-coated films were incubated in 1 mL of borate buffer containing 10 μL of 0.2 wt% solution of FITC in dimethyl sulfoxide for 2 h. The films were washed with deionized water and observed using the microscope.

### 2.4. Cell Adhesion

All cell experiments were performed using films sterilized by γ-radiation (0.2 Mrad).

To study the cell adhesion onto a surface of the initial and collagen-coated PLA films, we used human mesenchymal stromal cells (hMSCs) derived from adipose tissue biopsies after receiving a patient’s signed informed consent. The cells were isolated and expanded as described elsewhere [20]. To confirm their phenotype, we applied a microfluidic cell sorter, the Sony SH800 (Sony Biotechnology, Kyoto, Japan), using a standard antibodies’ panel previously described in [20,21]. The experiment was performed using the cells of passage 4. The sample films were seeded with 100,000 cells per cm^2^ and cultured using DMEM/F12 growth medium supplemented with 10% fetal calf serum (HyClone, Logan, UT, USA), gentamycin (50 μg/mL, Paneco, Moscow, Russia), L-glutamine (5 mg/mL, Gibco, MA, USA), basic fibroblast growth factor (20 ng/mL, ProSpec, Rehovot, Israel), and insulin-transferrin-sodium selenite (1:100, Prospec, Rehovot, Israel) under standard conditions (37 °C, 5% CO_2_). On Day 3, the samples were fixed with 3% glutaraldehyde solution in PBS for one hour at room temperature, then washed three times with PBS, and covered with 1% OsO_4_ solution in PBS for one hour with further triple washing with PBS. The samples’ dehydration was performed using ethanol (50%, 70%, 96%; twice for 5 min each) and acetone. They were then dried using a critical point dryer, covered with gold in vacuum, and analyzed using a CamScan-S2 scanning electron microscope (Cambridge Instruments, London, UK).

### 2.5. Cytotoxicity

To reveal the films’ cytotoxicity, we applied 3T3 mouse fibroblasts provided by the Biobank of the Sechenov University (Moscow, Russia). The contact cytotoxicity was studied using live/dead staining [22]. The samples were seeded with 100,000 cells per cm^2^ and cultured using DMEM/F12 growth medium supplemented with 10% fetal calf serum (HyClone, Logan, UT, USA) and 1% penicillin-streptomycin (Gibco, Waltham, MA, USA). On Day 3, they were stained with calcein-AM and propidium iodide and analyzed using a laser scanning confocal microscope LSM 880 equipped with an AiryScan module and GaAsP detector (Carl Zeiss, Oberkochen, Germany). The extract cytotoxicity was studied using lactate dehydrogenase (LDH-) and 3-(4,5-dimethylthiazol-2-yl)-2,5-diphenyltetrazolium bromide (MTT-) assays, as described elsewhere, in accordance with manufacturers’ instructions [22]. The films’ extracts (6 cm^2^) were prepared by dipping into 1 mL DMEM/F12 supplemented with 5% fetal calf serum (HyClone, Logan, UT, USA) and 1% penicillin-streptomycin (Gibco, Waltham, MA, USA) and placed for 24 h in a CO_2_-incubator. The cells were cultured in a 96-well plate (5000 cells per one well) and in 24 h treated with films’ extracts’ or sodium dodecyl sulfate (SDS) (positive control) serial dilutions.

## 3. Results and Discussion

### 3.1. Physicochemical Properties

Effectiveness of the chemical entrapment technique is based on semi-dissolution of the polymeric surface layer followed by adsorption of the targeted component from its solution and depends on the nature of the substrate/component, solvent mixture, and treatment time [9]. Previously we adopted the polylactide film pre-swelling procedure using a water/acetone mixture for successful entrapment of chitosan within the surface layer and used these conditions for collagen immobilization within the current work as well [5]. In contrast to the chemical entrapment method, the effectiveness of collagen immobilization via the plasma pre-treatment technique depends on surface characteristics of the plasma-modified polylactide substrate. Therefore, as a first step, the effect of DC-discharge modification of polylactide films on their surface chemical structure, morphology, and characteristics was evaluated.

In terms of future immobilization of collagen onto plasma-treated film, its surface chemical structure plays a key role. XPS data of surface layer analysis of the initial and plasma-treated PLA films at the anode and at the cathode are shown in Table 1. The oxygen concentration decreased after the treatment at the anode and slightly increased after the treatment at the cathode. A certain amount of nitrogen and aluminum was also found on the surface of the plasma-treated PLA films, which could be related to the nature of the working gas (air) and electrode etching, as was shown previously [12].

The high-resolution C1s/O1s/N1s core-level XPS spectra of the initial and plasma-treated PLA samples were analyzed to reveal the changes in the surface chemical composition as a function of the electrode used for the plasma treatment. Figure 1a shows the structural formula of PLA with marks of non-equivalent atoms and corresponding peaks in core-level XPS spectra. As can be seen from Figure 1 and Table 2, the C1s spectrum of the untreated PLA could be approximated by three peaks: the main one at 285.0 eV assigned to C−C/C−H, and the peaks at 286.9 and 289.1 eV related to C−O and O=C–O groups, respectively. O1s spectra of the initial PLA film showed two peaks typical for this polymer at 532.2 and 533.6 eV, attributed to O=C and O−C groups.

Plasma treatment leads to increase in the area of peak C-1 and a corresponding decrease in the areas of the peaks C-2 and C-3. This trend was more pronounced for the treatment at the anode. A significant increase in the half-width of the C-3 peak could indicate a partial transformation of the ester groups −COOC− into the carboxyl groups −COOH. These two groups produce C1s peaks with similar binding energies, resulting in a broadening of the C-3 peak. The clear two-component O1s spectrum in the initial sample was transformed into a wide peak at 532.5 eV in the spectra of the plasma-treated samples. The broadening of both O1s and C1s peaks after the plasma treatment could be related to formation of the carboxyl groups at the PLA film surface. Treatment of PLA by DC-discharge also generated a formation of nitrogen groups at 400.3 eV related to N−C bonds.

The type of the DC-discharge electrode used for the film treatment affects not only surface chemical composition, but morphology as well. As can be seen in Figure 2, plasma treatment of PLA film at the anode led to a drastic increase in surface roughness: the R_ms_ values of initial PLA film and PLA-A film were 0.26 and 2.04 nm, respectively. In contrast, the roughness of the film treated at the cathode was slightly lower (R_ms_ of PLA-C of 0.19 nm) in comparison with the initial PLA sample. This difference could be related to the nature of plasma active components affecting the surface: electrons possess smaller size and higher specific energy than ions and, therefore, could create a number of small deep holes as a result of surface etching by plasma.

The changes in the surface chemical structure and morphology led to increase of surface hydrophilicity. As can be seen in Table 3, the contact angle of wettability by water (θ_w_) dropped from 75° (non-treated PLA films) to almost a full water drop spreading (11–12°) in the case of DC-discharge treatment, irrespective of the electrode used. Calculation of the values of the work of adhesion, the total surface energy, and its components for plasma-treated PLA films showed a significant increase in surface energy in comparison with initial PLA film characteristics. Plasma treatment led to a 4.3–6.4-fold increase in the polar component of surface energy, which was more pronounced in the case of the PLA-C sample.

Surface hydrophilization and formation of polar groups onto plasma-treated films provides suitable conditions for successful immobilization of collagen. Since the collagen dip-coating was carried out immediately after plasma treatment, even a covalent bonds’ formation could be anticipated, but ionic interactions were also enough for collagen immobilization onto plasma-treated PLA substrate (Figure 3). In the alternative technique, the collagen deposition onto PLA films pre-swelled in solvent mixture (i.e., chemical deposition methods) is determined by physical entrapment of protein chains within the semi-dissolved polylactide surface layer. Taking into account the rather high Mw of both PLA and collagen, their mutual permeability could be hampered.

Immobilization of collagen onto plasma-treated PLA films led to a partial restoration of surface hydrophobicity (cf. Table 3 and Table 4). Thus, the contact angle of wettability (θ) of collagen-coated plasma-treated PLA films was 60° irrespective of the electrode used for the plasma pre-treatment. This effect could be a result of superposition of several events. First, the effect of surface hydrophilization of plasma-treated polymeric substrates is well-known to be impermanent due to rearrangement of the polar groups generated on the surface to the bulk phase of the polymer [23]. This process of re-arrangement normally requires more time than was spent on preliminary plasma treatment of PLA films, incubation in collagen solution, washing, drying, and θ measurements (approx. 7 h in total), but it could be accelerated by incubations in solution during the collagen deposition process. A second reason for partial surface hydrophobicity restoration after collagen immobilization could be related to removal of the surface charge observed immediately after plasma-treatment of PLA films [18]. Third, the θ value of the coated PLA film could be related to characteristics of the collagen coating itself. The measurements of θ of the model collagen film cast from its solution in acetic acid showed the same 60 degrees.

The contact angle of wettability of the PLA film coated with collagen via the chemical entrapment method (PLA-S-col sample) showed only a negligible improvement of surface hydrophilicity. The main changes in surface characteristics of this sample occurred in terms of the dispersive component of surface energy, which was doubled in comparison with the respective value of the initial PLA film. Since the dispersive component is particularly sensitive to surface morphology, a change in topography as a result of surface layer semi-dissolution and collagen entrapment could be the most likely reason for this effect.

Morphology of the initial PLA film as well as those coated with collagen via either chemical entrapment or immobilization onto plasma-treated films was studied using SEM. As can be seen in Figure 4, in contrast to the initial film, the PLA-S-col sample possessed more heterogeneous morphology, which could be caused by the various sensitivities of crystalline and amorphous regions of semi-crystalline PLA to swelling in solvent.

Fluorescent microscopy of the collagen-coated PLA films after their staining with selective-to-proteins fluorescein dye, FITC was used as a second method to confirm successful collagen deposition. As can be seen in Figure 5, all collagen-coated films demonstrated emission coming from the fluorescein dye bonded to protein. The native PLA films (without the collagen coating) showed no emission due to the absence of protein that could bind the FITC. The amount of collagen that could be immobilized onto PLA surface by these techniques is rather low, since it is mostly the protein adsorbed onto the activated substrate surface [5]. Taking into account the data on surface contact properties (Table 4) and fluorescent intensity (Figure 5) of the PLA films coated with collagen using different methods, it is possible to assume that plasma treatment allowed depositing more collagen. However, the synergetic effect of the substrate activation technique and the collagen coating on the cell adhesion and growth is more important for further biomedicine application.

### 3.2. Cell Adhesion and Cytotoxicity Analysis

The main aim with protein deposition was to improve the cell adhesion on a polylactide surface. Scanning electron and laser scanning confocal microscopy images showed that 3 days after seeding on the PLA film, cells remained viable, but were mostly rounded and poorly adhered to a surface (Figure 6). The cells cultivated on the collagen-coated films attached and migrated better, and were elongated and bipolar or multipolar. In general, the plasma treatment of the PLA films before collagen coating was shown to promote better cell adhesion and proliferation. The best results were observed for cells cultured on the PLA-A-col films. In accordance with live/dead staining, only few 3T3 cells cultured on all PLA film samples were revealed to be dead.

To analyze the extract cytotoxicity, we used two methods based on different cell functions: MTT assay—to measure cellular metabolic activity as an indicator of cell viability; LDH assay (LDH release assay)—to assess the level of plasma membrane damage as an indicator of cell death. The MTT assay showed no notable cytotoxicity of the prepared PLA films because the lowest cell viability detected was above 70%. The LDH release levels in 3T3 cells treated with the sample extracts were not higher than the spontaneous ones. The achieved results are similar to the previously reported data [24]. The combination of methods used permitted us to prove that the samples analyzed had no notable cytotoxicity.

The better cell adhesion of the plasma-treated films in comparison with the initial PLA film and the one coated with collagen via pre-swelling technique could be related to the effect of plasma treatment itself, as was shown earlier [12,18]. The parameters of plasma treatment could be varied in a wide range to optimize its effect on surface biocompatibility. In contrast, the pre-swelling technique is easier and does not require special equipment. An additional advantage of the chemical entrapment method is its applicability to different form of materials. In contrast, plasma treatment could not be so easily used for modification of the bulk of porous 3D materials or surfaces of powder particles without an aim of improved technology setups.

## 4. Conclusions

Plasma treatment is widely used and an effective technology to modify the surface of biocompatible materials and to activate them before coating deposition. This approach requires special equipment and it was compared with a more simple method of deposition of collagen onto polylactide films in order to enhance their biocompatibility with mammalian cells. Plasma treatment led to a significant effect on surface chemical structure, morphology, and hydrophilicity of the polylactide films. The subsequent immobilization of collagen onto plasma-treated polylactide films was followed by partial restoration of surface contact properties. The alternative approach to collagen immobilization onto polylactide films, chemical entrapment, was also successful and enhanced the adhesion of cells. The collagen-coated films pretreated by plasma showed the better in vitro results, but the ability to support cell adhesion and growth was mainly government by plasma treatment conditions rather than the presence of collagen coating.

## Figures and Tables

**Figure 1 polymers-14-04886-f001:**
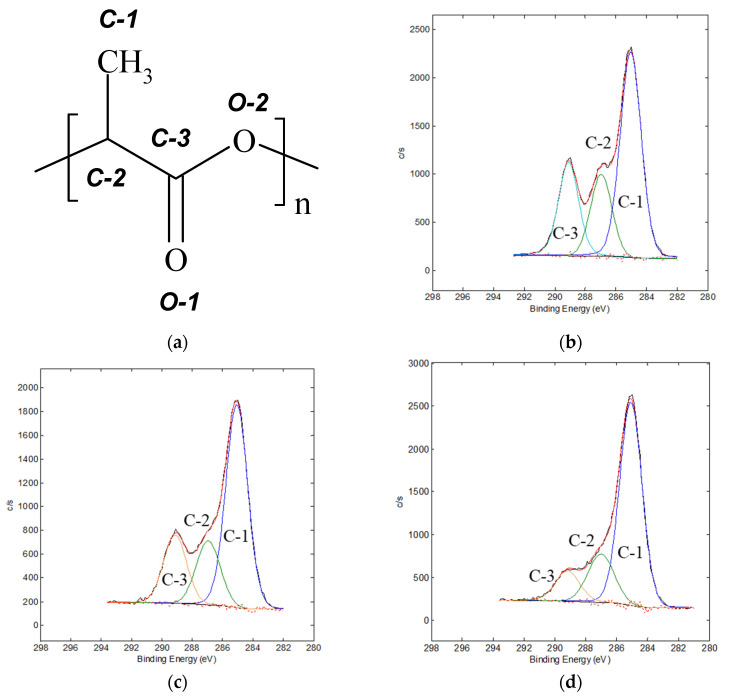
(**a**) PLA structural formula with peak marks and C1s spectra of the films, (**b**) initial, and (**c**) treated at the cathode, or (**d**) at the anode.

**Figure 2 polymers-14-04886-f002:**
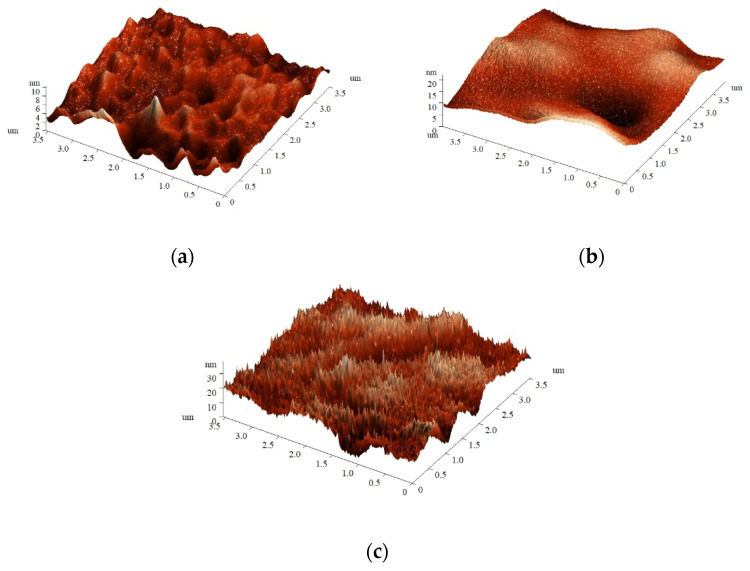
Atomic force microscopy images of (**a**) the initial PLA film and PLA films treated (**b**) at the cathode and (**c**) at the anode.

**Figure 3 polymers-14-04886-f003:**
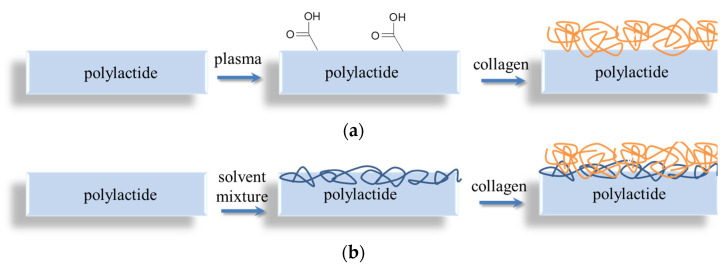
Schemes of collagen deposition onto PLA films via (**a**) plasma pre-treatment and (**b**) chemical entrapment methods.

**Figure 4 polymers-14-04886-f004:**
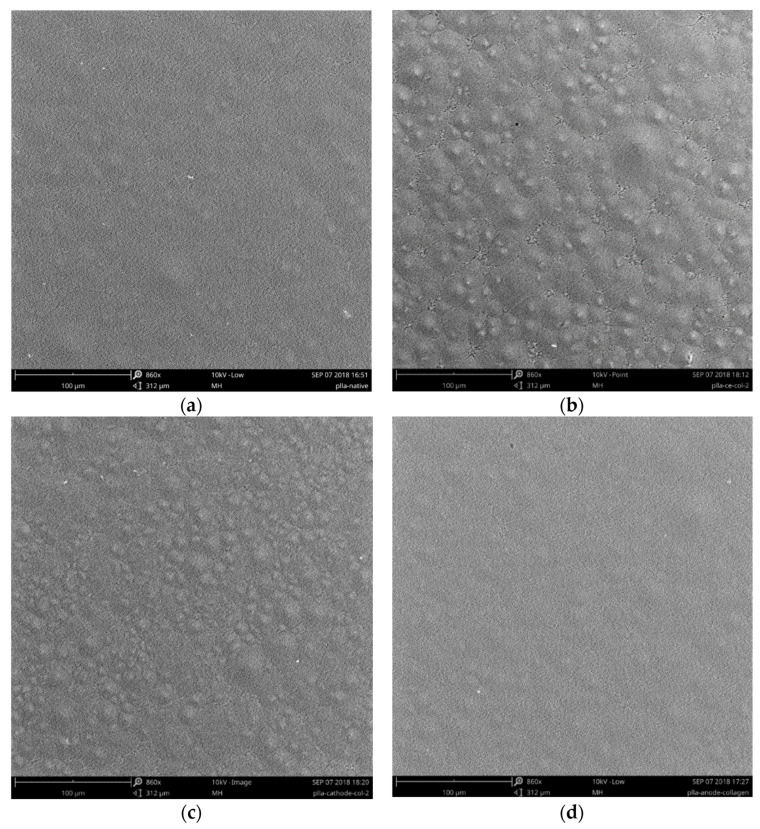
SEM micrographs of (**a**) the initial PLA films and the collagen-coated PLA films: (**b**) PLA-S-col, (**c**) PLA-C-col, (**d**) PLA-A-col. Scale bar is 100 μm.

**Figure 5 polymers-14-04886-f005:**
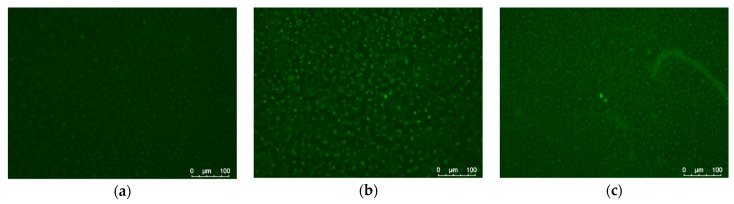
Fluorescent microscopy of FITC-labeled collagen-coated PLA films: (**a**) PLA-S-col, (**b**) PLA-C-col, (**c**) PLA-A-col.

**Figure 6 polymers-14-04886-f006:**
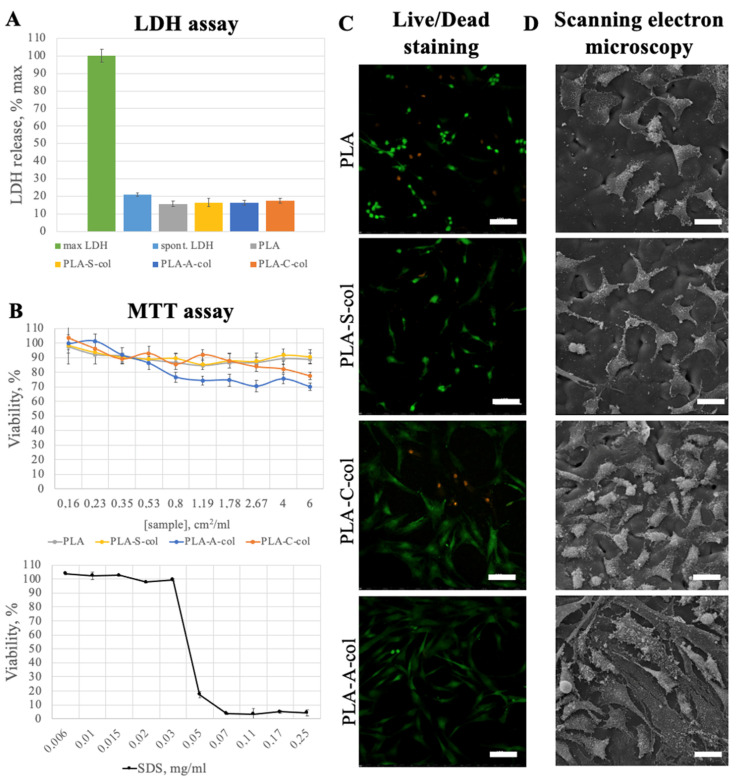
Cell adhesion and cytotoxicity results: (**A**) LDH assay (spontaneous LDH release (spont. LDH) and maximum LDH release (max LDH) were used as a control); (**B**) MTT assay (SDS was a positive control); (**C**) Live/Dead staining of 3T3 cells seeded on the PLA films (merged, calcein AM (green)—live cells, propidium iodide (red)—dead cells), 3 days of cultivation, laser scanning confocal microscopy, scale bar is 100 µm; (**D**) hMSCs seeded on the PLA films, 3 days of cultivation, scanning electron microscopy, scale bar is 30 µm.

**Table 1 polymers-14-04886-t001:** Surface chemical structure of the initial and plasma-treated PLA films.

Sample	Atomic Concentration, %				
	C	O	N	Al	O/C
PLA	67.0	33.0	-	-	0.49
PLA-C	61.5	34.2	2.0	2.3	0.56
PLA-A	72.5	23.8	2.5	1.2	0.33

**Table 2 polymers-14-04886-t002:** Approximation parameters of high-resolution XPS core-level spectra of the initial and plasma-treated PLA films.

Sample	Parameter	C1s			O1s		N1s
		C-1	C-2	C-3	O-1	O-2	
PLA	Binding energy, eV	285.0	286.9	289.1	532.2	533.6	−
	Peak half-width, eV	1.61	1.63	1.54	1.63	1.86	−
	area, %	55	21	24	46	54	−
PLA-C	Binding energy, eV	285.0	286.9	289.1	532.3		400.3
	Peak half-width, eV	1.71	1.87	1.79	−	−	−
	area, %	60	20	20	−	−	−
PLA-A	Binding energy, eV	285.0	286.9	289.1	532.7		400.4
	Peak half-width, eV	1.71	2.0	1.8	−	−	−
	area, %	70	19	11	−	−	−

**Table 3 polymers-14-04886-t003:** Surface contact properties of initial and plasma-treated PLA films.

Sample	θ, deg.		W_a_, mJ/m^2^		γ, mJ/m^2^		
	Water	Glycerol	Water	Glycerol	γ	γ^p^	γ^d^
PLA	75	71	91.6	84.0	29.0	18.0	11.0
PLA-C	11	56	144.3	98.9	115.3	114.5	0.8
PLA-A	12	37	144.0	114.0	81.7	78.1	3.6

**Table 4 polymers-14-04886-t004:** Surface contact properties of the initial and collagen-coated PLA films and the collagen film.

Sample	θ, deg.		W_a_, mJ/m^2^		γ, mJ/m^2^		
	Water	Glycerol	Water	Glycerol	γ	γ^p^	γ^d^
PLA	75	71	91.6	84.0	29.0	18.0	11.0
PLA-C-col	60	43	109.2	109.8	48.0	15.2	32.8
PLA-A-col	60	33	109.2	97.7	40.9	28.6	12.3
PLA-S-col	73	63	94.1	92.8	33.5	13.3	20.2
Collagen	60	56	109.5	113.5	52.8	12.3	40.6

## Data Availability

Data are contained within the article.

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
