# Peer review of "Effective and Easy Techniques of Collagen Deposition onto Polylactide Films: DC-Discharge Plasma Treatment vs. Chemical Entrapment"

_polymers, 2022, doi:10.3390/polym14224886_

Round 1

Reviewer 1 Report

Review of polymers-1987852

This is a nice manuscript about plasma treatment for attaching biomolecules onto polymer surface (a biodegradable polymer PLA, to be specific) that has considerable potential for biomedical materials applications. The results in this manuscript are properly characterized, by using XPS, AFM, MTT, LDH, SEM, especially with the surface energy analysis.

  1. Line 48-59: This section will be improved more with the addition of plasma polymerization technique that is missed to be covered in this manuscript. Please kindly refer to some articles about amine-containing plasma polymers for mammalian cells studies and biomedical material applications, as follows:

·       Biochem. Eng. J. 78 (2013) 198-204 https://doi.org/10.1016/j.bej.2013.02.022

·       J. Polym. Sci. Part B 51 (2013) 1361-1367 https://10.1002/polb.23341

  1. Line 75-76: This paragraph contains one sentence only. Please merge it with Line 71-74.
  2. Line 98-100: This paragraph contains one sentence only. Please merge it with Line 95-97.
  3. Line 120: …acetone. They were then…
  4. Line 126: …[25, 26]…
  5. Line 246-251: Please kindly add a short discussion or analysis about Figure 5b that has the most fluorescent intensity (has more collagen deposition compared to that of Figure 5c, and especially to that of Figure 5a, the darkest among all).
  6. Reference 6: Please revise the writing of the initial of the first name of the authors, some of them are repeated two or three times.

Author Response

This is a nice manuscript about plasma treatment for attaching biomolecules onto polymer surface (a biodegradable polymer PLA, to be specific) that has considerable potential for biomedical materials applications. The results in this manuscript are properly characterized, by using XPS, AFM, MTT, LDH, SEM, especially with the surface energy analysis.

Line 48-59: This section will be improved more with the addition of plasma polymerization technique that is missed to be covered in this manuscript. Please kindly refer to some articles about amine-containing plasma polymers for mammalian cells studies and biomedical material applications, as follows:

Biochem. Eng. J. 78 (2013) 198-204 https://doi.org/10.1016/j.bej.2013.02.022

  1. Polym. Sci. Part B 51 (2013) 1361-1367 https://10.1002/polb.23341

IEEE Trans. Plasma Sci. 49 (2021) 2428-2434 https://doi.org/10.1109/TPS.2021.3092820

Plasma treatment for biodegradable polymer and drug release application:

Appl. Surf. Sci. 262 (2012) 114-119 https://doi.org/10.1016/j.apsusc.2012.03.003

Answer: Thank you very much for the suggestion! We completely agree that this technique should be covered. The Introduction section was updated within the revised version of the manuscript.

Line 75-76: This paragraph contains one sentence only. Please merge it with Line 71-74.

Answer: The paragraphs were merged.

Line 98-100: This paragraph contains one sentence only. Please merge it with Line 95-97.

Answer: The paragraphs were merged.

Line 120: …acetone. They were then…

Answer: The sentence was corrected.

Line 126: …[25, 26]…

Answer: The references were corrected.

Line 246-251: Please kindly add a short discussion or analysis about Figure 5b that has the most fluorescent intensity (has more collagen deposition compared to that of Figure 5c, and especially to that of Figure 5a, the darkest among all).

Answer: Thank you for the comment! The amount of collagen, which could be immobilized onto the PLA surface by these techniques, is rather low, since it’s mostly the protein adsorption onto the activated substrate surface. Indeed, it seems that Figure 5a is the darkest among all, and taking into account the data on surface contact properties (Table 4) and fluorescent intensity (Figure 5) of the PLA films coated with collagen using different methods, it’s possible to assume that the plasma treatment allowed depositing more collagen. We added a short discussion on this matter within the revised version of the manuscript.

Reference 6: Please revise the writing of the initial of the first name of the authors, some of them are repeated two or three times.

Answer: The reference was corrected.

Reviewer 2 Report

This study set out to compare collagen coating of PLA films using plasma treatment and chemical entrapment methods. The introduction and purpose of the paper was clear.

In the cell adhesion and cytotoxicity sections, the area of the sample films was not defined. Also the method of preparation of the films for cell culture (sterilization and fixation in the cell culture dish) should be specified.

Is there a reason that the primary hMSCs weren't used for the cytotoxicity studies?

The method and results sections describing the FITC stain are worded strangely. I think the collagen was labeled with FITC, which binds to all proteins.

Is there a way to quantify the actual amount of collagen the adheres to the surfaces? Or some way to compare the effiency of the methods for trapping/binding collagen? If the point of the study is to directly compare the methods, this seems like a very relavent outcome.

The abbreviations LDH and MTT are not defined. And the assays are not discussed in a way that makes it clear why both were used. Also, the method of determining spontanious and max LDH values are not explained.

The order that the groups are presented in changes from figure to figure, which make it a little confusing to follow. Particularly in the LDH and MTT assays where the control PLA group is listed in the middle.

Is there any hypothesized reason that the two plasma treated groups seems to show more toxicity at the more concentrated dilutions? There is no acknowledgement in the discussion.

Author Response

This study set out to compare collagen coating of PLA films using plasma treatment and chemical entrapment methods. The introduction and purpose of the paper was clear.

In the cell adhesion and cytotoxicity sections, the area of the sample films was not defined. Also the method of preparation of the films for cell culture (sterilization and fixation in the cell culture dish) should be specified.

Answer: The information on area and sterilization was added. The films before cell seeding did not require to be fixed. To perform SEM, the fixation procedure is described as follows: “…On Day 3, the samples were fixed with 3% glutaraldehyde solution in PBS for one hour at room temperature, then washed three times with PBS, and covered with 1% OsO4 solution in PBS for one hour with further triple washing with PBS. The samples’ dehydration was performed using ethanol (50%, 70%, 96%; twice for 5 min each) and acetone. They were then dried using a critical point dryer, covered with gold in vacuum and analyzed using a CamScan-S2 scanning electron microscope (Cambridge Instruments, UK)…”

Is there a reason that the primary hMSCs weren't used for the cytotoxicity studies?

Answer: To perform the cytotoxicity analysis, we used 3T3 cell line in accordance with ISO 10993-5.

The method and results sections describing the FITC stain are worded strangely. I think the collagen was labeled with FITC, which binds to all proteins.

Answer: Thank you for the question! We modified the experimental section within the revised version of the manuscript. The native PLA film and collagen-coated PLA films were incubated in 1 mL of borate buffer containing 10 mL of 0.2 wt% solution of FITC in dimethyl sulfoxide for 2 hours. The films were several times washed with deionized water and observed using the fluorescent microscope. The collagen-coated PLA films showed an emission coming from fluorescein dye, which bonded to collagen. The native PLA films (without the collagen coating) showed no emission due to the absence of protein, which could bound the FITC.

Is there a way to quantify the actual amount of collagen the adheres to the surfaces? Or some way to compare the effiency of the methods for trapping/binding collagen? If the point of the study is to directly compare the methods, this seems like a very relavent outcome.

Answer: We considered several ways to compare the coating efficiency, such as energy-dispersive X-ray spectroscopy, X-ray photoelectron spectroscopy, FTIR-microscopy and weight measurements by microbalance. Some of them were tested within our previous papers describing chitosan coating deposition onto PLA or PET substrates [T.S. Demina, M.S. Piskarev, O.A. Romanova, A.K. Gatin, B.R. Senatulin, E.A. Skryleva, T.M. Zharikova, A.B. Gilman, A.A. Kuznetsov, T.A. Akopova, P.S. Timashev Plasma Treatment of Poly(ethylene terephthalate) Films and Chitosan Deposition: DC- vs. AC-Discharge // Materials 2020, 13(3), 508; https://doi.org/10.3390/ma13030508; T.S. Demina, A.А. Frolova, A.V. Istomin, S.L. Kotova, M.S. Piskarev, K.N. Bardakova, M.Yu. Yablokov, V.A. Altynov, L.I. Kravets, A.B. Gilman, T.A. Akopova, P.S. Timashev Coating of polylactide films by chitosan: comparison of methods // Journal of Applied Polymer Science. 2020. V. 137. № 3. https://doi.org/10.1002/app.48287]. Here, the amount of immobilized collagen is very low, due to it’s mostly the protein adsorption onto the activated substrate surface. Taking into account the data on surface contact properties (Table 4) and fluorescent intensity (Figure 5) of the PLA films coated with collagen using different methods, it’s possible to assume that plasma pre-treatment allowed depositing more collagen. We updated this section in the revised version of the manuscript. It seems to us that the synergetic effect of the substrate activation and collagen coating on the cell adhesion and growth is more important for biomedicine application.

The abbreviations LDH and MTT are not defined. And the assays are not discussed in a way that makes it clear why both were used. Also, the method of determining spontanious and max LDH values are not explained.

Answer: The relevant sections were expanded.

The order that the groups are presented in changes from figure to figure, which make it a little confusing to follow. Particularly in the LDH and MTT assays where the control PLA group is listed in the middle.

Answer: Figure 6 was corrected.

Is there any hypothesized reason that the two plasma treated groups seems to show more toxicity at the more concentrated dilutions? There is no acknowledgement in the discussion.

Answer: The observed difference in MTT-assay is statistically insignificant. The overall cytotoxicity measured by both methods (MTT- and LDH-assay) is less than 30 %; therefore, in accordance with ISO 10993-5, we could claim that the prepared PLA films showed no notable cytotoxicity.

Round 2

Reviewer 1 Report

Review of polymers-1987852-v2

The manuscript has been revised well. It can be accepted in its current form.

Note: Line 109 must be combined with line 110-126. This issue can be corrected during the proofreading stage. Thank you.